# Characterizing Extratropical Tropopause Bimodality and its Relationship to the Occurrence of Double Tropopauses Using COSMIC GPS Radio Occultation Observations

**Benjamin Johnston** [1,2,*] and Feiqin Xie [1]

1    Department of Physical and Environmental Sciences, Texas A&M University–Corpus Christi,
     Corpus Christi, TX 78412, USA; Feiqin.Xie@tamucc.edu
2    University Corporation for Atmospheric Research, Boulder, CO 80301, USA
*    Correspondence: bjohnston@ucar.edu

**Abstract:** Lapse rate tropopause (LRT) heights in the extratropics have been shown to display a bimodal distribution, with one modal maxima above 15 km (typical of the tropical tropopause) and the other below 13 km (typical of the extratropical tropopause). The climatology of the tropopause is studied by characterizing tropopause bimodality and how it relates to the occurrence of double tropopauses (DTs). LRT heights are derived from Constellation Observing System for Meteorology, Ionosphere, and Climate (COSMIC) GPS Radio Occultation temperature profiles from 2006 to 2017. Tropopause bimodality occurs most frequently within a subtropical band (20°–40°) in both hemispheres. A distinct seasonality is observed as bimodality occurs most frequently in winter except for another local maximum along the northern edge of the Asian summer monsoon. The regions with a bimodal height distribution nearly overlap the regions that experience a high frequency of DTs. DTs occur most frequently in winter (50%–70% of the time) along the poleward edge of the bimodal band, and most LRT heights are within the extratropical mode (>80%), whereas DT occurrence decreases quickly toward the equatorward edge (<20%) along with fewer LRT heights in the extratropical mode (<50%). These results indicate that LRT height bimodality occurs along the equatorward edge due to the occurrences of double tropopauses, while the poleward edge is due to single tropopause profiles that are more tropical in nature.

**Keywords:** extratropics; lapse rate tropopause; bimodality; double tropopauses; GPS radio occultation

## 1. Introduction

The tropopause is one of the most important locations of the atmosphere as it separates the turbulently mixed troposphere and the more stable stratosphere [1], and these two layers of the atmosphere are chemically, dynamically, and thermally distinct [2]. A fundamental characteristic of the tropopause is the change in static stability across the interface [3]. The World Meteorological Organization (WMO) definition of the tropopause is based on thermal criteria using the temperature lapse rate (lapse rate tropopause, or LRT), which provides a convenient way to define the tropopause [4]. The tropopause can also be defined by more general stability criteria using the potential vorticity-based dynamical tropopause [5] or the ozone-based chemical tropopause [6]. Characteristics of the tropopause and the surrounding region, the upper troposphere and lower stratosphere (UTLS), are important because the tropopause acts as a "two-way gate" for the exchange of mass, water vapor, and chemical species between the troposphere and stratosphere [7]. Specifically, stratosphere–troposphere exchange (STE) of ozone and water vapor has attracted substantial interest because these two trace gases

contribute significantly to atmospheric radiative forcing, and their distributions and controlling mechanisms are important elements of chemistry–climate interaction [8]. The long-term variability and trends of the tropopause height have been recognized as a climate change indicator [9,10] because of its location in a region of minimum temperatures and its sensitivity to changes in the concentration of radiatively active species in the UTLS [11,12]. For example, Santer et al. [13] showed that tropopause height is closely associated with tropospheric warming and stratospheric cooling. Thus, continuous monitoring of the global tropopause could enhance understanding of tropopause behavior and contribute knowledge to a variety of critical topics in atmospheric and climate research [3,14].

The boundary between the tropics and extratropics is often characterized by a split in the tropopause rather than a smooth transition, which results in discontinuities observed near the subtropical jet. Along this boundary, considerable variation in tropopause height has been observed in previous studies. For example, Seidel and Randel [15] showed that tropopause heights can display a bimodal structure throughout the subtropics using radiosonde data. In these regions, the tropopause is sometimes at the height of the higher tropical tropopause and other times at a height typical of the lower extratropical tropopause, which results in a clear separation of about 3–5 km between the two modes. Even though tropopause heights in the subtropics tend to be higher in summer and lower in winter, the authors pointed out that bimodality is not simply a reflection of seasonal variability, as high and low tropopauses were observed in all seasons at each location. However, one of the main restrictions of the study is the limited spatial coverage of the radiosonde data, which results in observations at only a few locations over land. Additionally, seasonal variation of tropopause heights was not analyzed in depth within the study. The location and strength of the subtropical jet is known to vary considerably with season, which could have an impact on when and where tropopause bimodality occurs.

Additionally, along the subtropical jet, the occurrence of multiple tropopauses have been found in atmospheric temperature profiles using radiosondes [1,16] and satellite data [17,18]. First, it is important to note the differences between tropopause bimodality and multiple tropopauses, as these terms sound relatively similar. For example, the occurrence of tropopause bimodality is determined by the statistical classification of a *group* of tropopause heights observed in a region. This classification is intended to separate tropical and extratropical tropopause heights and is typically observed using histograms or probability distribution functions. However, the occurrence of multiple tropopauses is a phenomenon whenever two or more tropopauses are observed within a *single* temperature profile, typically using the WMO definition of the LRT. Much research has been done over the past few decades to better characterize the thermal structure within multiple tropopause profiles. For example, it has been shown that a large percentage of temperature profiles display a double tropopause during the midlatitude winter (40%–80%), and this frequency decreases into the summer [18,19]. Early studies about the synoptic structure of double tropopause environments showed that they are related to upper-level jet stream frontal zones [20], while later studies have related them to ozone variability [6] and tropopause folding [1,19]. Randel et al. [3] showed that double tropopauses occur when the low latitude (tropical) tropopause extends to higher latitudes, overlying the lower tropopause. Additionally, they form most frequently above strong cyclonic circulation systems, as this results in reduced stability in the lower stratosphere. However, while tropopause bimodality and frequent occurrences of double tropopauses have been shown to occur throughout the subtropics, no clear relationship has been established in the literature up to this point.

GPS Radio Occultation (RO) offers a unique opportunity to observe tropopause structure and variability with accurate, high vertical resolution temperature soundings [21]. For example, Rieckh et al. [22] used decade-long GPS RO data to show that variability in the subtropical tropopause break leads to a large spread in the tropopause height and temperature distribution in the 20° to 30° latitudinal band during winter. Additionally, they noted that the median tropopause height is systematically higher than the mean at these latitudes, since most tropopause heights were at higher altitudes (e.g., tropical) with some also showing much lower altitudes (e.g., extratropical). The authors

also analyzed double tropopauses and showed that they display weak latitudinal variations, with most second tropopause heights between 14 and 20 km in the extratropical winter hemisphere and between 18 and 20 km in the tropics. Son et al. [23] also found interesting intraseasonal variabilities in tropopause properties using COSMIC GPS RO measurements. For example, tropopause temperature and pressure show significant variability in the subtropics along the subtropical jet in both hemispheres, the storm track regions in the winter hemisphere, and the Tibetan plateau in the summer. Zhang et al. [24] also found that the largest seasonal variations of tropopause heights occur in the subtropics, with variations of 2 to 4 km observed within 30°–40° in both hemispheres based on GPS RO climatological data.

The purpose of this study is to better understand the climatological characteristics of the extratropical tropopause by examining tropopause bimodality and establishing how it relates to double tropopause occurrence. The three main questions we seek to answer in this manuscript are as follows. (1) Where and when does tropopause bimodality occur? Tropopause bimodality has been demonstrated in some extratropical radiosonde observations [15], but a long-term global record of this phenomenon is still absent. (2) What are the characteristics of the tropopause in the locations where bimodality occurs? Answering this question will provide information about the climatology of the subtropical jet stream and how often each region observes different synoptic patterns. (3) How does the occurrence of double tropopauses relate to bimodality? Double tropopauses occur frequently throughout the subtropics [1,3,17–19] where tropopause bimodality has been identified to occur. Therefore, answering this question could provide additional understanding about bimodal tropopause formation mechanisms and characteristics. In this study, we use COSMIC GPS RO soundings to identify tropopause heights, as COSMIC offers a relatively long data record of global observations of UTLS temperatures and tropopause heights with a high vertical resolution. The structure of the paper is as follows: Section 2 provides background on the GPS RO data and the methodology used in this study, including tropopause definitions and how tropopause bimodality is characterized; Section 3 presents the key results of the study, including bimodal tropopause locations and characteristics and how they relate to double tropopause formation; Section 4 provides a brief discussion of the results; lastly, the conclusions are provided in Section 5.

## 2. Data and Methods

The atmospheric refractivity $N$, a key observable of GPS RO, depends on conditions in the dry atmosphere and water vapor [25]:

$$N = 77.6\frac{P}{T} + 3.73 \times 10^5 \frac{e}{T^2} \qquad (1)$$

where $P$ is the atmospheric pressure (in hPa), $T$ is the temperature (in K), and $e$ is the partial pressure of water vapor (in hPa) [26]. The dry temperature is derived from the refractivity (1) by neglecting atmospheric humidity [27] such that:

$$T_{dry} = 77.6\left(\frac{P_{dry}}{N}\right) \qquad (2)$$

where $P_{dry}$ is the dry pressure (i.e., the pressure without water vapor) derived through hydrostatic integration [28]. In this study, dry temperature is used because it can be treated as an independent satellite retrieval, whereas the real temperature retrieval relies on a priori moisture information from the European Centre for Medium-Range Weather Forecasts (ECMWF) low-resolution operational analysis. Note that the dry temperature retrieval is nearly identical to the real temperature in the UTLS region as moisture is negligible [28]. To ensure that this is the case, we only consider dry temperatures <240 K when identifying the tropopause [29].

GPS RO soundings are obtained from the joint US–Taiwan six-satellite FORMOSAT-3/COSMIC (FORMOSA Satellite Series No. 3/Constellation Observing System for Meteorology, Ionosphere,

and Climate) mission [21]. The COSMIC constellation provided up to 2500 soundings per day shortly after launch in 2006 with sampling coverage provided around the globe. However, as individual satellites within the constellation have gone offline throughout the study period, the number of daily soundings has decreased to roughly 250–300 per day at the end of 2017. We obtained the reprocessed level-2 RO profiles from CDAAC (COSMIC Data Analysis and Archive Center) at the University Corporation for Atmospheric Research (UCAR). The profiles are quality controlled by excluding the ones with "bad" flags (such as if the observation bending angles exceed the climatology by a specific threshold). We use the "atmPrf" product, which provides refractivity and dry temperature from usually near the surface up to approximately 60 km. The vertical resolution of RO soundings varies from 0.2 km in the lower troposphere to 1.4 km in the upper stratosphere [21] with an average of 0.5 km in the UTLS. The retrieved profiles are reported as a function of geometric height above mean sea level (orthometric height; not to be confused with geopotential height), and the location of each profile nearest to the surface is used.

To obtain extratropical tropopause climatological characteristics, COSMIC GPS RO temperature profiles are analyzed from 2006 to 2017. The thermal tropopause height is determined by the WMO lapse rate tropopause (LRT) definition [4]: (1) The first tropopause is defined as the lowest level at which the temperature lapse rate decreases to 2 K km$^{-1}$ or less, provided that the average lapse rate between this level and all higher levels within 2 km does not exceed 2 K km$^{-1}$; (2) If above the first tropopause, the average lapse rate between any level and all higher levels within 1 km exceeds 3 K km$^{-1}$, then a second tropopause can be defined by the same criterion as (1). Additionally, our tropopause-detection algorithm does not begin searching for the tropopause height until the height of the profile is 5 km above mean sea level to alleviate issues with dry temperature profiles in the lower troposphere. Each GPS profile is interpolated into a 10 m uniform vertical grid using a quadratic interpolation scheme and then smoothed to 500 m, which is roughly the native resolution for RO profiles in the UTLS altitude range. The GPS profiles are binned into 2.5° latitude × 5° longitude grids and then the median background tropopause characteristics (e.g., LRT height and temperature) for four seasons are derived (December–January–February, March–April–May, June–July–August, September–October–November). The number of temperature profiles in each seasonal grid generally ranges from 250 to 350 profiles throughout the extratropics.

Tropopause bimodality is evaluated using Otsu's method [30], which is an automatic image thresholding technique that creates a binary image (e.g., monochrome) based on setting a threshold value on the pixel intensity of the original image (e.g., grayscale) to separate the foreground pixels from background pixels [31,32]. This method searches for the threshold that maximizes the inter-class variance of a histogram, which is defined as the weighted sum of variances of the two classes:

$$\sigma^2 = \omega_0 \omega_1 (\mu_1 - \mu_0)^2 \tag{3}$$

where $\omega$ are class weights, $\mu$ are class means, and the subscripts represent the two groups within the histogram. The method involves iterating through all the possible threshold values between each histogram bin and calculating a measure of spread for the pixel levels on each side of the threshold (so that the pixels fall in either the foreground or the background). The location that maximizes the inter-class variance of the histogram then separates the pixels into two classifications. Otsu's method can also be applied to other cases of unsupervised classification besides image thresholding in which a histogram is available of some feature (such as tropopause heights) that is discriminative for classification [30]. Otsu's method demonstrates good performance if the histogram displays a relatively deep valley separating two peaks. However, the performance of this technique can be limited by small sampling, the small mean difference between the two classes, and large amounts of noise [33].

In this study, histograms of LRT height distribution are generated for each grid in every season. Then, Otsu's method is applied to determine whether the tropopause displays a bimodal distribution in that grid location. Figure 1 displays the LRT height histograms and inter-class variances for December–January–February (DJF) and June–July–August (JJA) for a grid over Florida (27.5°–30°N,

80°–85°W). In DJF (Figure 1a), tropopause heights display a well-defined bimodal distribution, with two clear modes separated by a deep valley. The bimodal threshold, or the height separating the two modes, occurs at 15 km. The sum of the relative frequencies of the two groups of LRT heights show similar values (both near 50%) although the second mode with higher LRT heights shows a considerably higher peak frequency with less spread. Additionally, an extremely large variety of LRT heights are observed during the wintertime, with heights ranging from 8.5 km up to 19 km. This indicates a mixture of environments occurring, as LRT heights > 15 km (i.e., the upper mode) are characteristic of a tropical environment and LRT heights < 13 km (i.e., the lower mode) are typical for an extratropical environment [15]. The corresponding maximum variance (Figure 1c) is approximately 4.5 and occurs at 15 km. In contrast, the histogram for JJA (Figure 1b) displays a near-normal distribution, and the tropopause is consistently high (tropical). There are much smaller LRT height variations displayed, which results in a small maximum variance (Figure 1d) of approximately 0.45.

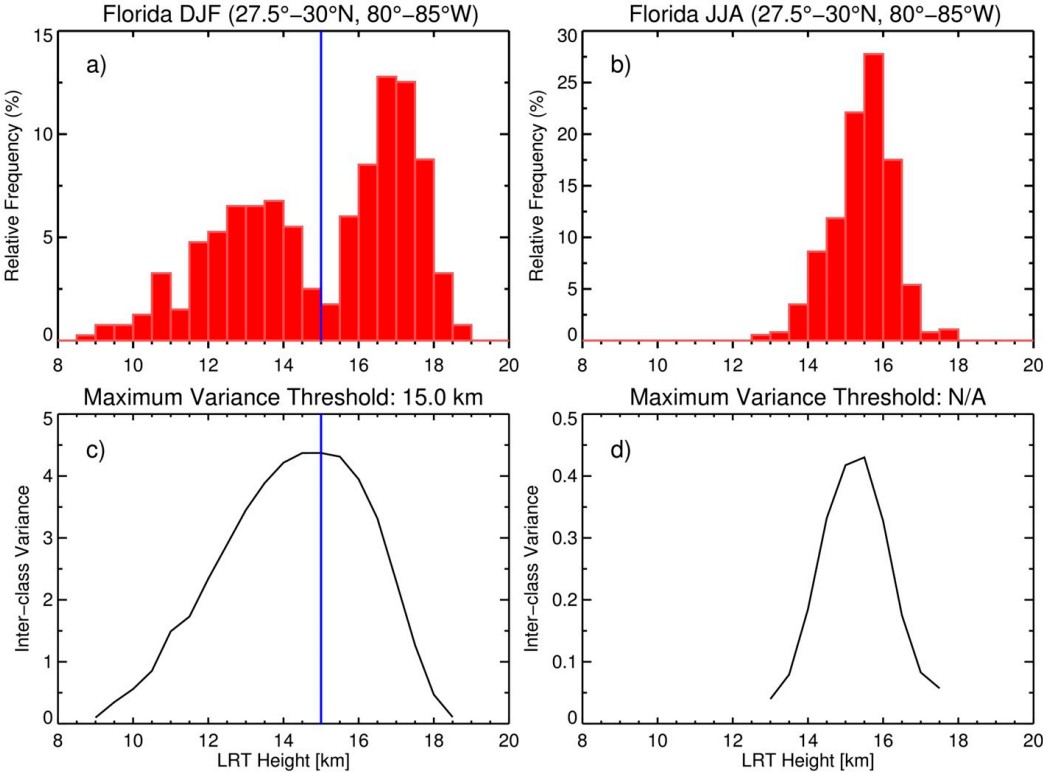

**Figure 1.** Lapse rate tropopause (LRT) height histograms and inter-class variances in (**a**,**c**) December–January–February (DJF) and (**b**,**d**) June–July–August (JJA) for a grid located over Florida (27.5°–30°N, 80°–85°W). Blue line indicates the height (the bimodal threshold) separating the two modes if the distribution is bimodal.

Additionally, careful analysis of a considerable number of tropopause height histograms around the globe throughout every season revealed that additional constraints must be implemented to ensure the tropopause heights are properly classified as bimodal. This is necessary because Otsu's detection algorithm assumes that the input histogram always has a bimodal distribution. However, most locations around the globe do not experience tropopause bimodality. Therefore, two additional criteria are required to assure robust detection of LRT height bimodality, such as (1) the sum of the relative frequency of occurrences for each class must be at least 10%, and (2) the maximum inter-class variance of the histogram must be at least 2.5. For example, over the high latitude North Atlantic Ocean in winter, relatively large maximum inter-class variances (approximately 2) are observed due to a wide variety of observed LRT heights. However, the LRT height histograms display a near-normal distribution.

## 3. Results

### *3.1. Bimodal Tropopause Locations and Characteristics*

This section examines the bimodality of the extratropical tropopause by determining where and when bimodality occurs and what the characteristics are of each mode. Seidel and Randel [15] (hereafter SR2007) used radiosonde data from 1979 to 2005 along with collocated NCEP/NCAR reanalysis data to demonstrate the occurrence of bimodal LRT heights at four different subtropical radiosonde stations. However, their analysis does not address the seasonal variation of bimodality. Figure 2 displays the seasonal probability density functions (PDFs) of COSMIC LRT heights at four grids close to the SR2007 radiosonde stations. The LRT heights are binned using 1-km intervals from 5 to 20 km. A distinct seasonality in LRT heights is observed in each region. In Beit Dagan (Figure 2a), bimodality occurs in DJF and March–April–May (MAM) with very similar PDFs, as there are two well-defined peaks with deep valleys in between. However, bimodality is not evident in JJA and September–October–November (SON). Large peaks occur above 15 km (characteristic of the tropical tropopause) during these seasons with less frequent occurrences of the lower tropopause heights. For Kashi (Figure 2b), seasonality is also observed but with opposite characteristics. Bimodality is now only observed in JJA and SON, with JJA displaying the larger peak for the higher (tropical) tropopause heights, whereas the larger peak during SON is observed for the lower (extratropical) heights. In contrast, heights throughout DJF and MAM are almost exclusively of the extratropical variety. This difference between Beit Dagan and Kashi can be attributed to the differences in latitude, as Kashi is much farther north and experiences extratropical heights more frequently. At Perth (Figure 2c), bimodality is observed in three seasons (MAM, JJA, and SON). In JJA, the total occurrence frequency of each mode is very similar, whereas MAM and SON are skewed toward the tropical heights. The results at Durban (Figure 2d) are comparable, as bimodality is also observed in MAM, JJA, and SON with similar modal characteristics displayed. COSMIC LRT heights show considerable seasonal variability for the occurrence of bimodality. Each location experiences large differences in the occurrence frequency of the tropical and extratropical mode, and these variations occur due to seasonal variations in the location and strength of the subtropical jet [18].

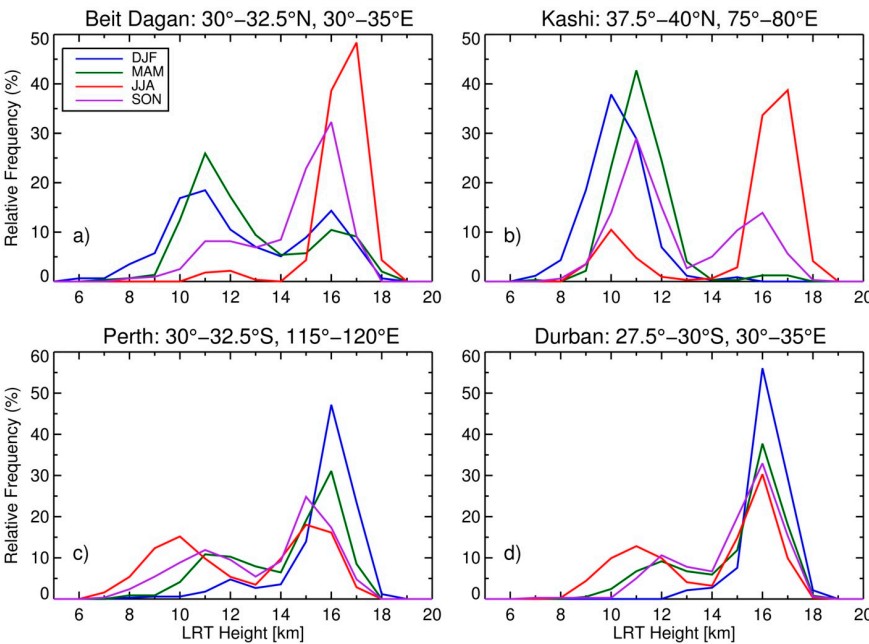

**Figure 2.** Probability distribution functions (PDFs) of Constellation Observing System for Meteorology, Ionosphere, and Climate (COSMIC) LRT heights (km) in DJF (blue), March–April–May (MAM) (green), JJA (red), and September–October–November (SON) (purple) over (**a**) Beit Dagan, (**b**) Kashi, (**c**) Perth, and (**d**) Durban.

The next set of figures expands upon bimodal tropopause features by showing when/where tropopause bimodality occurs globally and what the characteristics are. Figure 3 maps the seasonal locations of bimodal tropopauses along with the altitude that separates the two modes (the bimodal threshold). In general, bimodality occurs more frequently during the winter due to a stronger subtropical jet and temperature contrasts [3]. In DJF (Figure 3a), bimodality occurs in the Northern Hemisphere continuously throughout, on average, an approximately 10°–20° wide band between 20° and 40° with considerable zonal variations due to the location of the subtropical jet. Bimodality in the Southern Hemisphere generally displays a width of approximately 10° and demonstrates much less meridional movement, except off the western coast of South America. The bimodal threshold is generally highest in DJF, with many locations on the equatorward side of the band experiencing threshold values above 14.5 km. This occurs because the tropical tropopause heights are typically highest in DJF (approximately 17 km) and lowest in JJA (approximately 16 km) [24]. Substantial bimodal threshold differences occur in the Northern Hemisphere DJF, with ranges reaching 3 km or more between the northern and southern edges of the band in the Northern Pacific. In MAM (Figure 3b), the Northern Hemisphere bimodal band begins to shrink in extent throughout the Western Hemisphere, with most of the occurrence relegated to the North American continent along with threshold values remaining high (≥14.5 km). However, occurrence remains similar to DJF throughout the Eastern Hemisphere or even increases in extent over the Asian continent. The Southern Hemisphere bimodal band characteristics are almost identical to DJF except for a slight shift equatorward (approximately 5°) and a reduction in the tropical "tail" off the west coast of South America. In JJA (Figure 3c), the only remaining bimodal region throughout the Northern Hemisphere is over continental Europe and Asia. This region is much farther north (generally between 40° and 50°) and is likely associated with the Asian monsoon circulation. The Asian monsoon circulation includes a strong anticyclonic flow in the UTLS, and double tropopauses frequently occur on the poleward flank of this anticyclonic circulation (the region of westerly zonal winds), where the tropical tropopause (near 16 km) extends poleward to almost 60° [3]. As a result, bimodal thresholds remain relatively high (mainly between 13.5 and 14.49 km) even at these higher latitudes. In the Southern Hemisphere winter, the zonal extent of bimodality remains relatively similar to the other seasons, although the bimodal threshold is generally lower. In general, bimodality occurs more frequently in the Northern Hemisphere winter, similar to the double tropopause frequencies shown in Peevey et al. [18]. This is likely due to hemispheric differences in land–ocean distribution [19]. The number of large mountain ranges in the Northern Hemisphere enhances the propagation of large wavenumber 2 waves, which can influence the development of double tropopauses [18]. In SON (Figure 3d), the Northern Hemisphere bimodal extent begins to increase, especially throughout the Western Hemisphere (such as over the northern United States). The location continues to remain at relatively higher latitudes (exclusively poleward of 30° and up to almost 50°), and height thresholds are dependent on latitude.

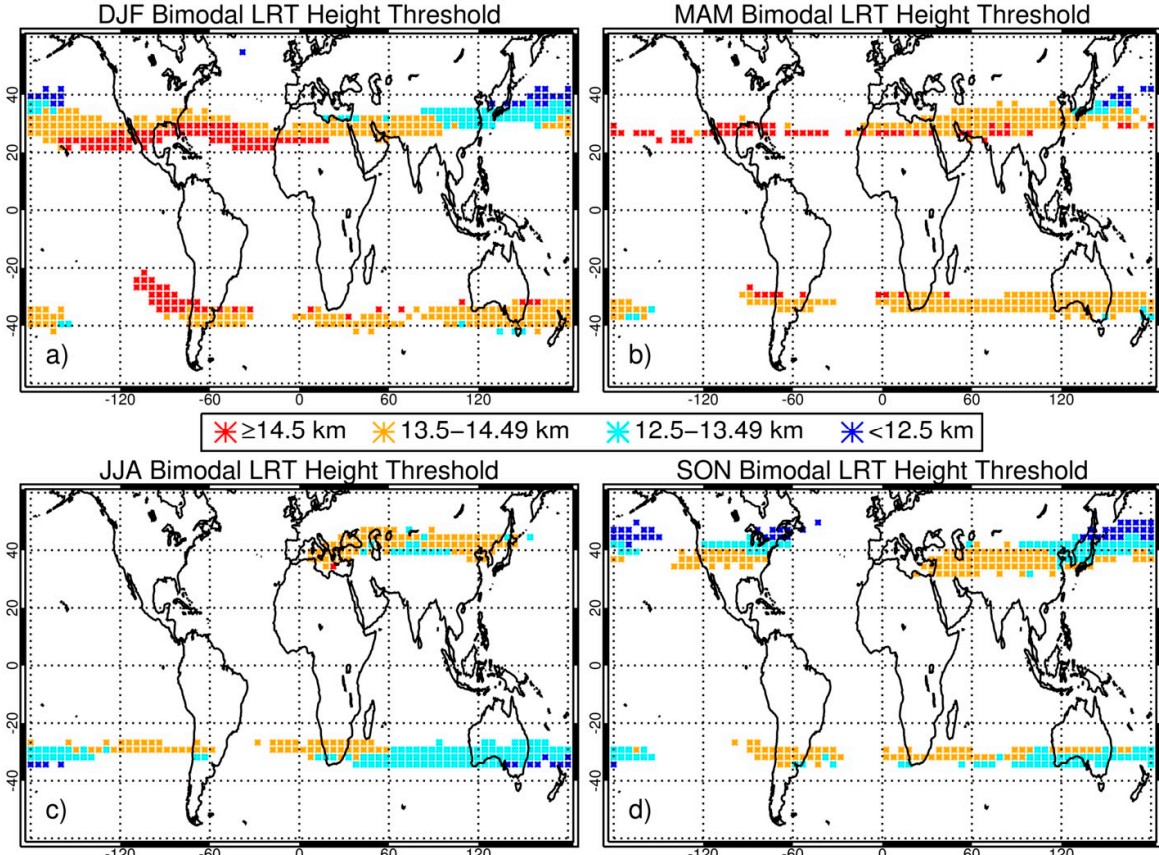

**Figure 3.** Maps of bimodal LRT height locations color-coded by the bimodal threshold height (km) in (**a**) DJF, (**b**) MAM, (**c**) JJA, and (**d**) SON.

Within each grid that observes bimodal LRT heights shown from Figure 3, the occurrence frequency for COSMIC profiles with LRT heights within the lower mode is shown in Figure 4. As expected, the modal frequency of occurrence displays a distinct shift within each band of bimodal LRT heights within all seasons, as the equatorward side of the band typically has fewer profiles with LRT heights within the lower mode (<40% occurrence), whereas the poleward side has more within the lower mode (>60%). This shift occurs relatively quickly (over the course of 2.5°–5° latitude) due to the characteristic "tropopause break" that occurs at this latitude, as the tropopause dips rapidly from the tropics to the extratropics. Double tropopauses occur frequently around this region as well. These features are also associated with the characteristic break in the LRT near the subtropical jet where the tropical tropopause extends to higher latitudes, overlying the lower extratropical tropopause [3]. Thus, the observed LRT bimodality, especially along the equatorward side of the bimodal region, may be related to these double tropopause environments and is investigated further in the next section.

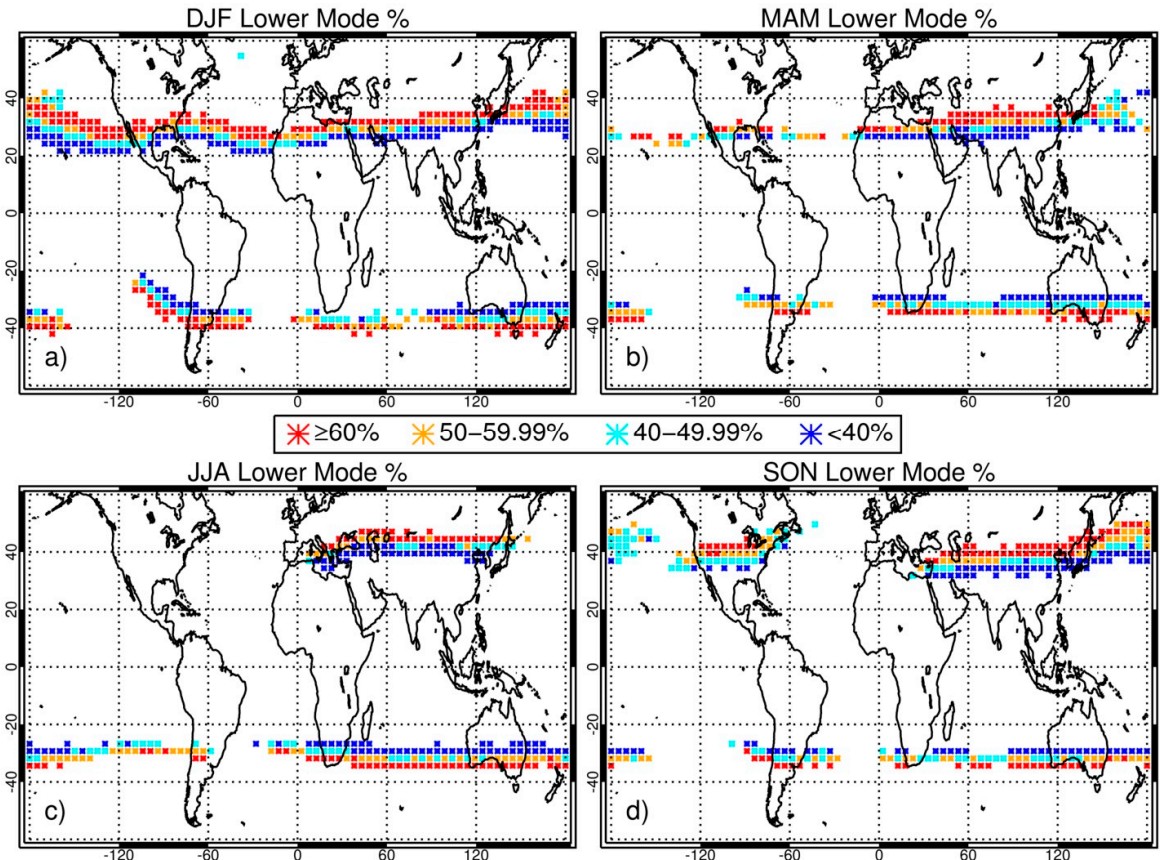

**Figure 4.** Maps of bimodal LRT height locations color-coded by the occurrence frequency (%) of COSMIC profiles with LRT heights in the lower mode in (**a**) DJF, (**b**) MAM, (**c**) JJA, and (**d**) SON.

### 3.2. Relationship of Double Tropopauses to the Occurrence of LRT Bimodality

This section will investigate the relationship between double tropopauses and bimodality. First, it is beneficial to demonstrate typical DT temperature profiles to understand the types of synoptic environments present. Figure 5 shows individual profiles with DTs from the same four locations discussed in Figure 2 (Beit Dagan, Kashi, Perth, and Durban) during specific seasons whenever bimodality occurs. There are two types of profiles shown for each location: the blue profile is when the first tropopause height (LRT1) is below the bimodal threshold, and the red profile is when it is above the threshold. In Beit Dagan (Figure 5a) and Perth (Figure 5c), the profiles are obtained from the winter, while the profiles for Durban (Figure 5d) are obtained from the fall. For all three locations, the two profiles look considerably different. The DT profiles with LRT1 heights in the lower mode (LM) all have LRT heights below 12 km. A strong inversion occurs just above LRT1 and is known as the tropopause inversion layer, which is a region of enhanced static stability above the extratropical tropopause associated with a narrow-scale temperature inversion [34]. This layer can be as shallow as 1 km (such as near Durban) or as thick as 4 km (such as near Beit Dagan). Then, above this layer, the temperatures begin to decrease again (albeit at a different lapse rate compared to below the first tropopause) until reaching a second tropopause near the typical tropical tropopause height (e.g., 17 km near Durban). In contrast, the DT profiles with tropopause heights in the upper mode (UM) display a much warmer upper troposphere (up to 5 K). However, temperatures quickly become colder above the LM profile's first tropopause, and the UM profile's tropopause is generally 5–10 K colder. This indicates that two different types of environments are present (tropical versus extratropical). Additionally, two summer profiles are shown for Kashi (Figure 5b) that look very similar, but with large differences in the strength of the inversion around 10–11 km. The inversion is strong and well-defined in the LM profile but is not strong enough in the UM profile to be classified as a tropopause according to the WMO definition.

We speculate that these differences could be due to the previously discussed Asian monsoon circulation. The strength of the anticyclonic circulation and associated westerly winds likely influence the type of profile observed, as a stronger circulation would produce a stronger inversion (and vice versa).

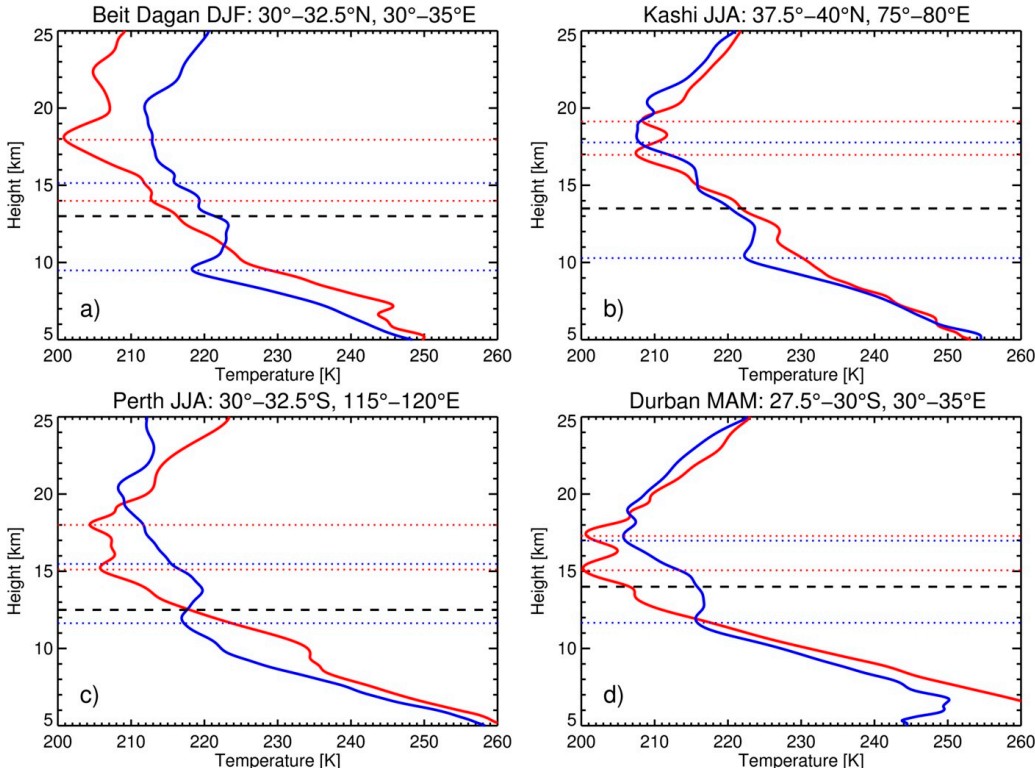

**Figure 5.** Typical double tropopause profiles when the LRT1 height is below (blue) and above (red) the bimodal threshold from four respective regions of (**a**) Beit Dagan, (**b**) Kashi, (**c**) Perth, and (**d**) Durban. The blue and red dotted lines are the locations of the first and second LRT heights, while the black dashed line is the bimodal threshold.

It is necessary to display double tropopauses characteristics to understand how these events relate to bimodal tropopause height distributions. Figure 6 shows the seasonal double tropopause frequency of occurrences along with the locations that display bimodal LRT heights. Double tropopause frequencies are generally highest in the winter in each hemisphere, as NH DJF (Figure 6a) shows a large area of frequencies >50% (with some values between 60% and 70%) along the poleward edge of the bimodal region, and SH JJA (Figure 6c) also displays a few areas with >50% occurrence frequencies. These values decrease slowly through the seasons, with NH minimum frequencies occurring in SON and SH minimum frequencies occurring in MAM (generally <35%). Additionally, frequencies decrease rapidly toward the equatorward edge of where bimodality occurs, as many locations observe frequencies <20%. Overall, the COSMIC RO locations of DTs agree well with previous research, although our frequencies are slightly lower than those found by Schmidt et al. [19], who saw maximum frequencies over 80% in some locations, and slightly higher than those of Peevey et al. [18], who generally saw frequencies up to 50%. Both studies relate the high frequency of DTs in the subtropics to the zonal mean wind speeds at 200 hPa, as the maximum zonal wind at this altitude serves as a rough indicator for the mean location of the jet streams [18,19]. The bimodal region is generally close to the regions that experience a high frequency of DTs. However, the regions do not display a perfect overlap, as the highest DT frequencies often continue poleward beyond where bimodality was identified for up to 5°–10°.

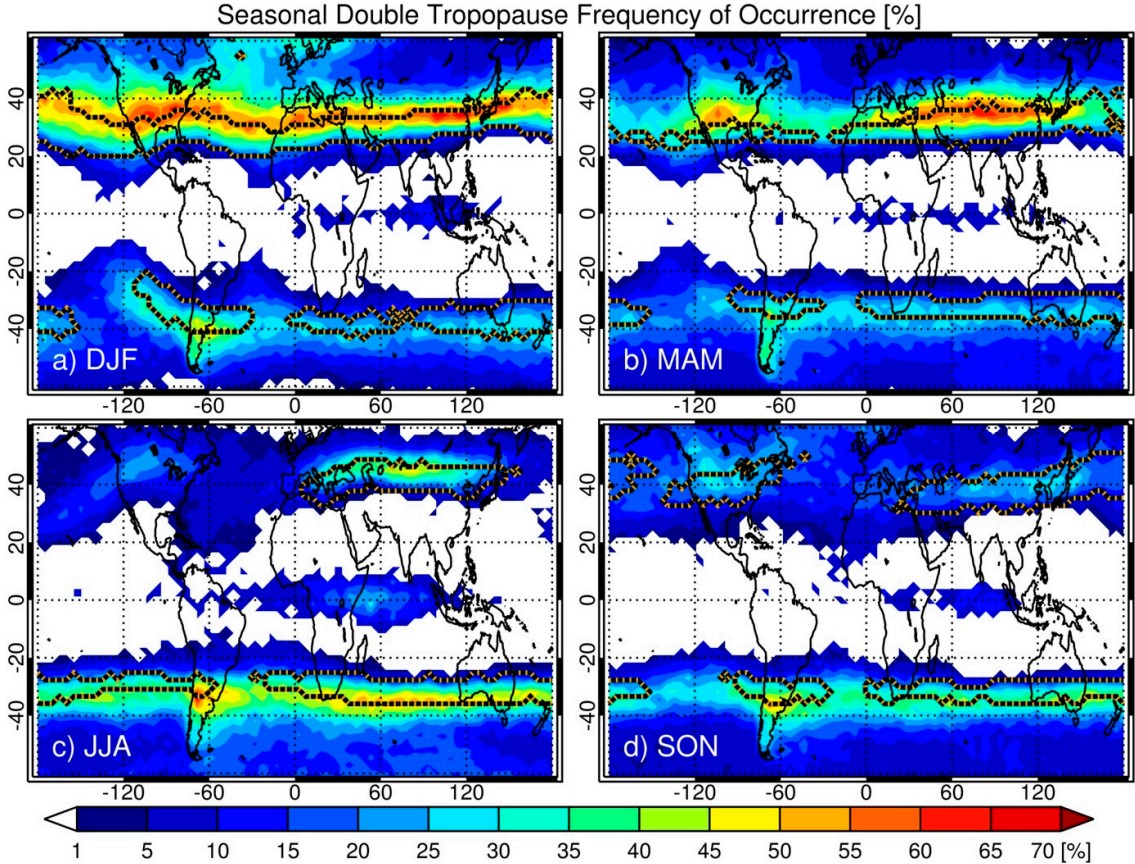

**Figure 6.** The occurrence frequency (%) of double tropopauses observed from COSMIC profiles in (**a**) DJF, (**b**) MAM, (**c**) JJA, and (**d**) SON. The region that displays LRT height bimodality is outlined by the black and gold dashed lines.

It is also useful to establish the type of environments most frequently associated with the formation of DTs and how the LRT1 forms in these profiles. Figure 7 shows the percentage of DT profiles that have their LRT1 height within that grid's lower mode. Similar to Figure 6, there is a distinct meridional shift in the percentage of profiles that have their LRT1 within the lower mode. For example, in DJF (Figure 7a), DT profiles on the poleward edge of the NH bimodal band almost always have a lower (extratropical) LRT1 height, as most grids display values of >80% (and some percentages are as high as 95%). In contrast, many grids on the equatorward edge of the bimodal band observe percentages of <50%. This indicates that the LRT1 for DT profiles at these latitudes can also occur at altitudes consistent with the tropical tropopause. Additionally, there are strong zonal asymmetries in the Northern Hemisphere, as latitudes such as 30° show considerable longitudinal structure (e.g., percentages in Southeastern Asia are <50% compared to the Eastern Pacific percentages of >80%). This same pattern is evident in every Northern Hemisphere season. Less variation is evident in the Southern Hemisphere as most grids observe frequencies of >65%, except in JJA (winter) and SON (spring). This result shows that not all DT environments are the same. While most DTs do produce LRT1 heights of extratropical origin (in the lower mode), sometimes DT profiles can have LRT1 heights within the upper mode. Even though most locations on the equatorward side of the bimodal band have smaller DT occurrence frequencies (generally <35%), these profiles would play a considerable role in producing tropopause bimodality in these locations.

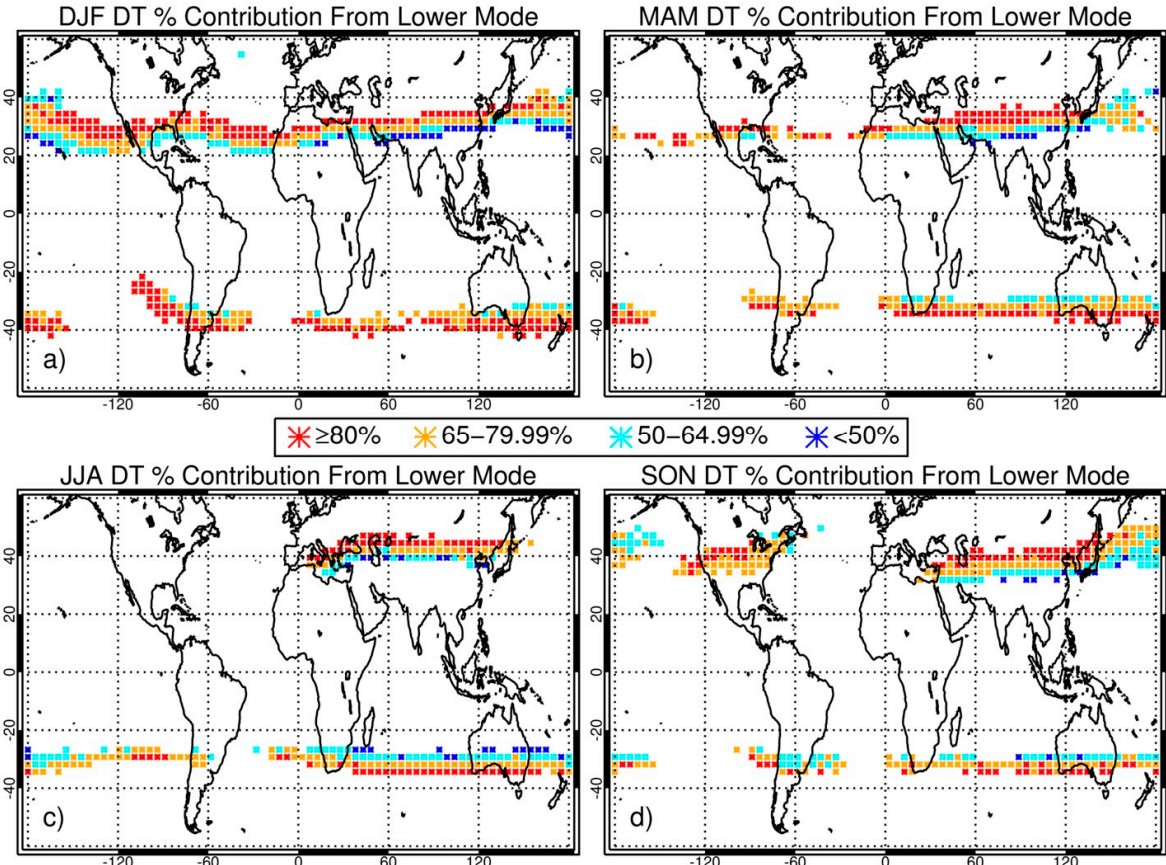

**Figure 7.** Maps of bimodal LRT height locations color-coded by the percentage of double tropopause profiles with LRT1 heights in the lower mode in (**a**) DJF, (**b**) MAM, (**c**) JJA, and (**d**) SON.

The previous figure indicates that not all temperature profiles with double tropopauses are identical. Even though most DT profiles do have their LRT1 within the lower mode (e.g., of extratropical origin), there are still some with their LRT1 within the upper mode. Characterizing the different structures of these two types of DT profiles is important in analyzing their synoptic environments and how DTs contribute to bimodal LRT height distributions. Figure 8 shows the gridded seasonal mean temperature profiles for double tropopause profiles with LRT1 heights within the lower mode (solid lines) and the upper mode (dashed lines). Two Northern Hemisphere and two Southern Hemisphere grids were chosen based on bimodality occurring throughout most or all of the year to allow for more seasonal comparisons. Bimodality does not occur year-round in any Northern Hemisphere grids (so locations with only three bimodal seasons, sans JJA, were chosen), whereas it does occur year-round in a few Southern Hemisphere grids. Two different types of double tropopause synoptic environments are apparent for each location. In Figure 8a, profiles with lower mode DT profiles look very similar throughout all seasons, with LRT1 heights generally between 11 and 12 km and second tropopause (LRT2) heights near 17–18 km. In contrast, the upper mode DT profiles in this region look more like single tropopause tropical profiles as the finer scale features are smoothed out in the mean profiles, but little variation is evident again between the seasons. The lower mode profiles are generally 5–10 K colder throughout the upper troposphere but generally become approximately 5 K warmer above the lower mode profile's LRT1. More seasonal variation is evident in Figure 8b, as both the lower mode and upper mode mean profiles display large temperature differences at all altitudes along with LRT height locations. Both Southern Hemisphere locations (Figure 8c,d) display profile characteristics that look relatively similar. For example, the LM and UM mean profile temperature magnitude, LRT1 heights, and profile shape are almost the same in each season. In summary, these profiles provide additional evidence that there are considerable differences among DT profiles depending on their LRT1

altitude, and these differences provide insight as to which large-scale environment is influencing the location at each time (tropical versus extratropical).

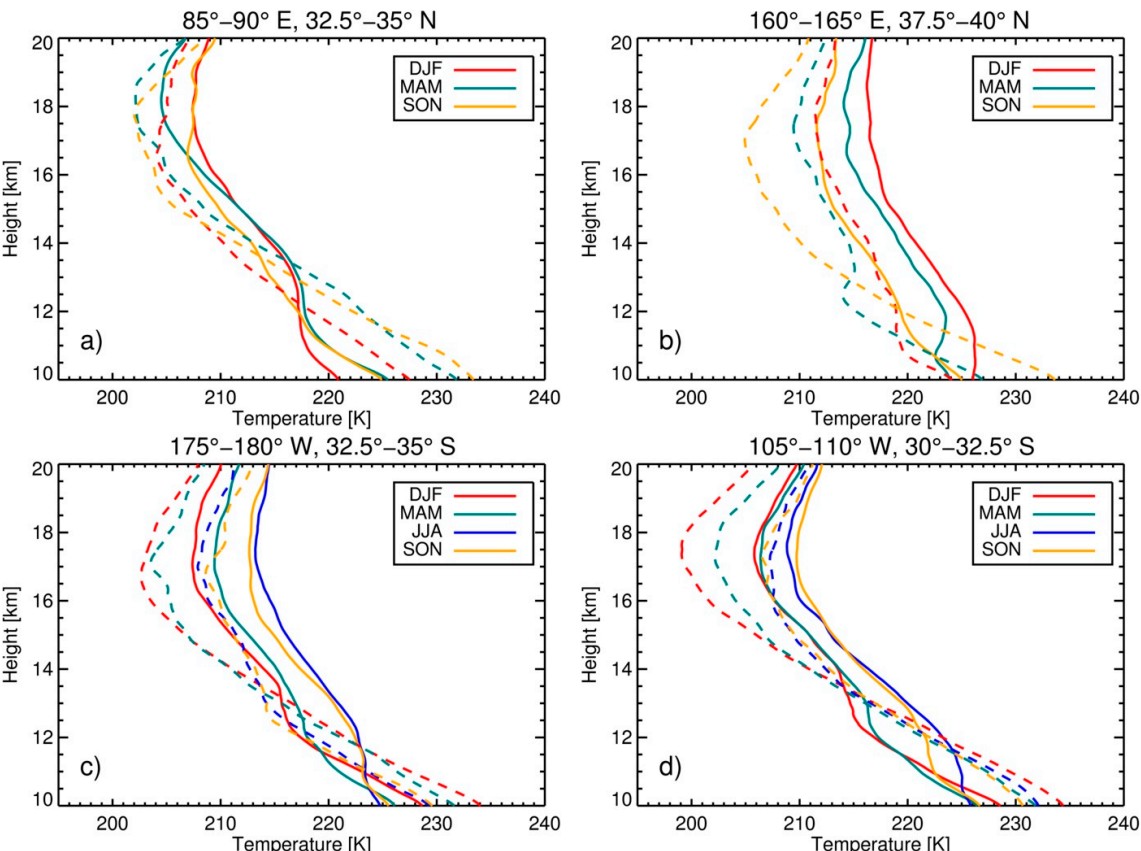

**Figure 8.** Seasonal mean temperature profiles within (**a**) 32.5°-35°N and 85°-90°E, (**b**) 37.5°-40°N and 160°-165°E, (**c**) 32.5°-35°S and 175°-180°W, and (**d**) 30°-32.5°S and 105°-110°W for double tropopause profiles with LRT1 heights in the lower mode (solid lines) and the upper mode (dashed lines) for DJF (red), MAM (teal), JJA (blue), and SON (orange).

Furthermore, COSMIC profiles in the study region are classified by whether they have single or double tropopauses in order to establish how the LRT height bimodality changes for each type of profile and help determine which type of profile plays a larger role in producing bimodality in different locations. Bimodality is shown for DT profiles in Figure 9 along with the bimodal threshold height. In general, the bimodal band is slightly reduced in extent relative to the results shown for all profiles (e.g., Figure 3), although in some locations, its occurrence is eliminated almost completely (such as over North America in SON). Bimodality generally occurs toward the equatorward side of the previously identified bimodal region, as now large portions of bimodality between 30° and 40° are not observed. The bimodal threshold is generally lower for DT grids in locations that overlap with bimodal grids observed from all profiles, as the threshold is typically one category lower (e.g., up to 1 km lower). This indicates that DT profiles often result in lower bimodal thresholds, since their LRT1 heights are typically lower than the LRT1 height of single-tropopause profiles, even when both profiles are extratropical in nature.

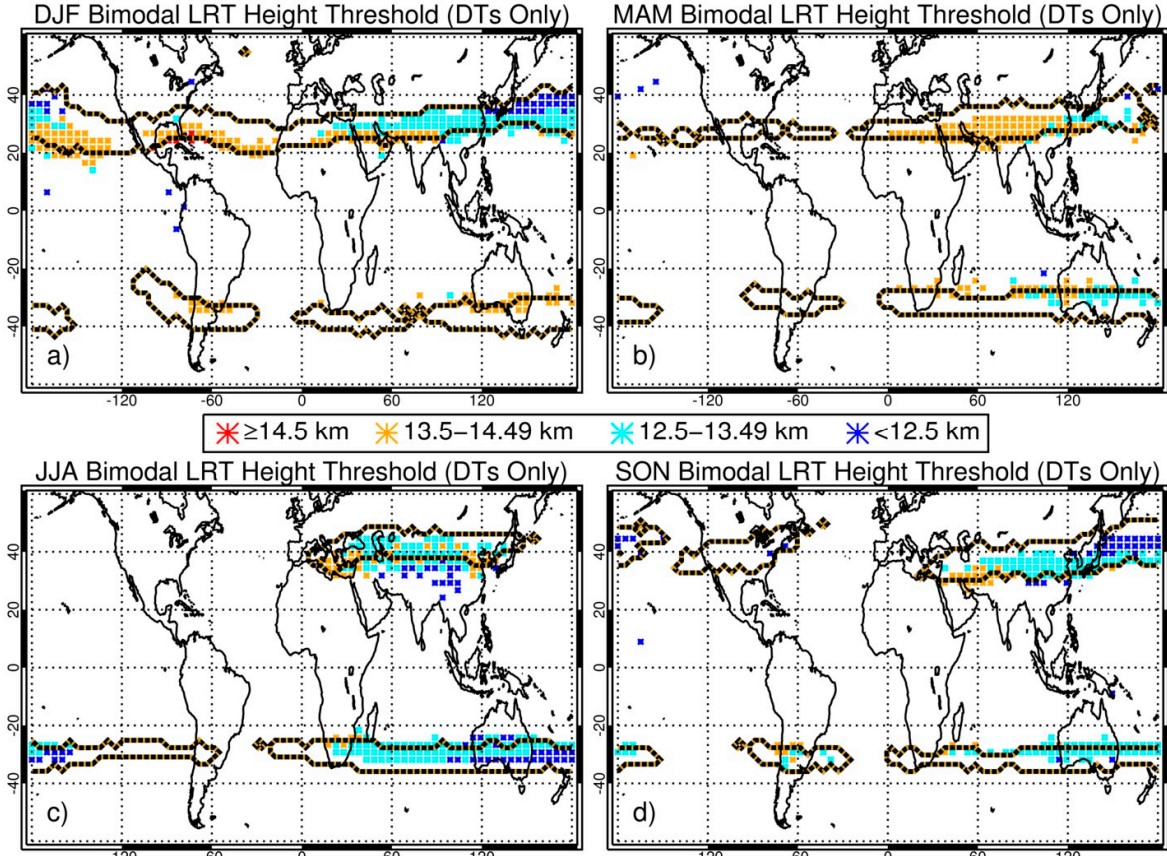

**Figure 9.** Maps of bimodal LRT height locations color-coded by the bimodal threshold height (km) only for COSMIC profiles that have a double tropopause identified in (**a**) DJF, (**b**) MAM, (**c**) JJA, and (**d**) SON. The region that displays LRT height bimodality is outlined by the black and gold dashed lines.

Finally, we determine where tropopause bimodality occurs when considering temperature profiles that only have a single tropopause identified along with the bimodal threshold height in Figure 10. The bimodal band becomes much narrower when examining single tropopause profiles, as its maximum width is generally only 2.5°–7.5° and shows many additional gaps. The location is shifted toward the poleward edge of the previously identified bimodal region throughout all seasons. For example, in DJF (Figure 10a), the NH band is confined between 30° and 40° whereas before, many locations also observed bimodality between 20° and 30°. This shift toward the poleward edge continues for all seasons and both hemispheres, and in some locations, the single tropopause bimodal band shifts poleward slightly beyond the bimodal band displayed for all profiles. At the same time, the bimodal threshold is generally higher for single tropopause-only grids in locations that overlap with bimodal grids from all profiles, as the threshold is often one category higher (e.g., up to 1 km).

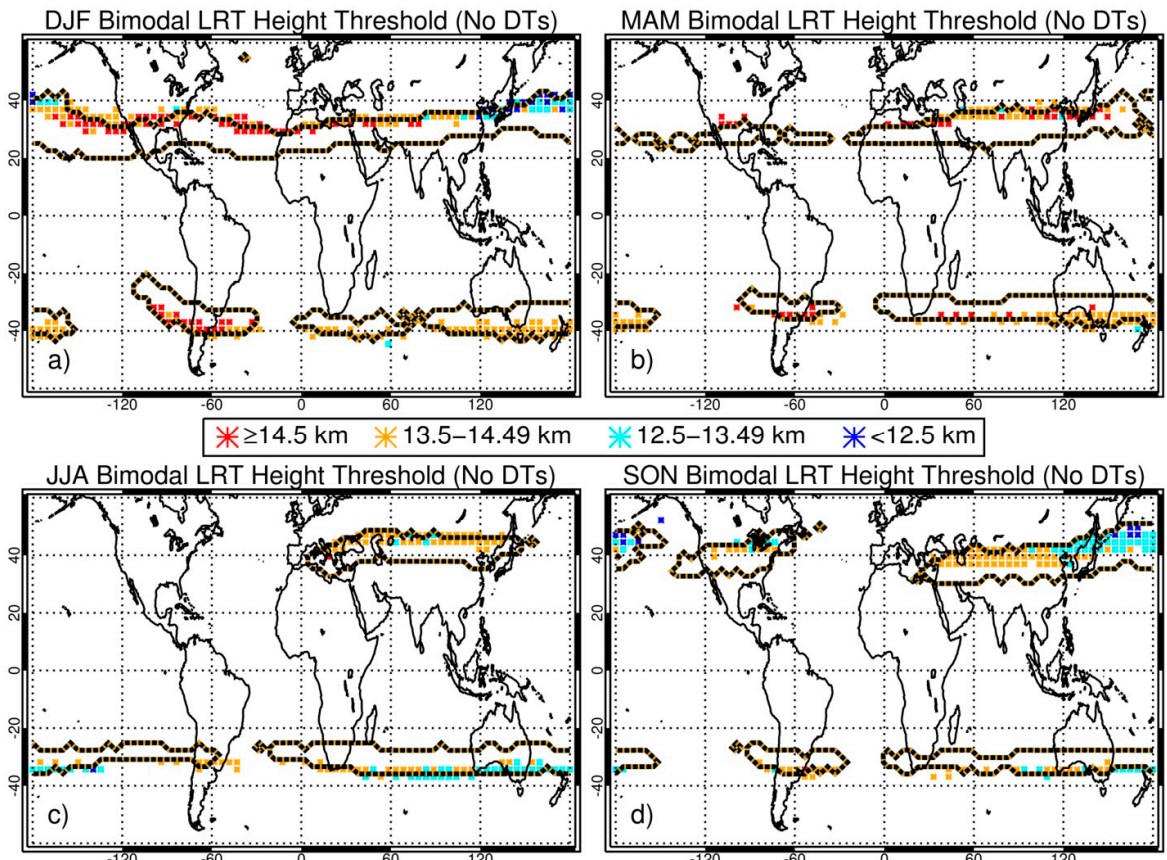

**Figure 10.** Maps of bimodal LRT height locations color-coded by the bimodal threshold height (km) only for COSMIC profiles that have a single tropopause identified in (**a**) DJF, (**b**) MAM, (**c**) JJA, and (**d**) SON. The region that displays LRT height bimodality is outlined by the black and gold dashed lines.

## 4. Discussion

These results suggest that two main factors contribute to the occurrence of seasonal LRT bimodality. On the equatorward side of the bimodal band, bimodality is mainly due to the occurrences of double tropopauses. At these latitudes, profiles with only a single tropopause are almost always more tropical in nature, as LRT heights are higher and occur in the upper mode (hence no bimodality shown in Figure 10). However, when double tropopause profiles occur at these latitudes (which is typically <30% of the time), their LRT1 heights are often occurring within the lower mode (between approximately 40% and 65% of the time), which results in bimodality. On the other hand, on the poleward side of the bimodal band, bimodality occurs due to single tropopause profiles that are more tropical in nature. At these latitudes, double tropopauses occur very frequently (often >50% of the time) with their LRT1 height mainly within the lower mode (80% to 95% of the time). Thus, if only double tropopause profiles are considered, most of these locations would not be classified as bimodal. However, the additional single tropopause profiles over the region have a mixture of lower (extratropical) and upper (tropical) mode LRT heights, creating the bimodality seen at these latitudes in Figure 10.

## 5. Conclusions

In this paper, climatological characteristics of the extratropical tropopause were explored using COSMIC GPS RO profiles by analyzing LRT bimodality and establishing its relationship to double tropopause occurrences. COSMIC soundings over regions near the SR2007 radiosonde stations were studied (Figure 2). A distinct seasonality in LRT height bimodality was observed, as each location experienced large variability in the occurrence frequency of the tropical (LRT > 15 km) and extratropical

(LRT < 13 km) mode. Furthermore, bimodality occurred in the Northern Hemisphere within 20°–50° latitude in a 10°–20° wide band with considerable zonal structure (Figure 3). Bimodality occurred more frequently during the winter, which we speculate is due to a stronger subtropical jet and larger temperature contrasts [3] and shrank in extent throughout spring and into summer as temperature contrasts weaken. However, bimodality persisted in the summer over northern Asia, which is likely due to the Asian monsoon circulation. Bimodality in the Southern Hemisphere generally displayed a width of approximately 10° latitude in all seasons and demonstrated much less meridional movement. In general, bimodal thresholds remained relatively high (>13.5 km) on the equatorward side of the band and decreased quickly on the poleward side of the band (<12.5 km) except in the Northern Hemisphere summer.

The relationship between double tropopauses and bimodality was also investigated. The location of the bimodal region was generally close to the locations that experience a high frequency of DTs (Figure 6). DT frequencies were highest in the winter (50% to 70%) along the poleward edge of the bimodal region, and these frequencies generally decreased steadily through the seasons and toward the equatorward side of the bimodal band (20% to 30%). Previous studies [18,19] demonstrated a strong relationship between the high frequency of DTs in the subtropics to the zonal mean wind speeds at 200 hPa, as the maximum zonal wind speeds at this altitude serves as an indicator for the mean location of the jet streams. Gridded seasonal mean temperature profiles for DT profiles with LRT1 heights within the lower mode and upper mode were shown (Figure 8) in regions that experience bimodality throughout most or all of the year. Considerable differences were observed between the mean profiles at each location depending on whether the LRT1 was within the upper or lower mode, and these differences provided more evidence as to which large-scale environment was present (tropical or extratropical) and why bimodality was occurring.

Further analysis (Figures 9 and 10) suggests that two main factors contribute to the occurrence of seasonal LRT bimodality. On the equatorward side of the bimodal band, bimodality is mainly due to the occurrence of double tropopauses. At these latitudes, profiles with a single tropopause are almost always tropical in nature with LRT1 heights in the upper mode. However, when double tropopause profiles occur in this region, their LRT1 heights often occur within the lower mode, which results in bimodality. On the other hand, on the poleward side of the bimodal band, bimodality occurs due to single tropopause profiles that are more tropical in nature. At these latitudes, double tropopauses occur very frequently, and their LRT1 height is almost always within the lower mode. Thus, if only double tropopause profiles are considered, most of these locations would not be classified as bimodal. However, single tropopause profiles have a mixture of extratropical and tropical LRT1 heights, which results in the observed bimodality. Further research on this topic is planned in the future as more RO data becomes available. A longer-term trend analysis of how LRT bimodality is changing could provide a better understanding of tropopause behavior and interactions between the tropics and extratropics.

**Author Contributions:** Conceptualization, B.J. and F.X.; methodology, B.J and F.X.; software, B.J.; validation, B.J.; formal analysis, B.J.; investigation, B.J.; resources, F.X.; data curation, B.J.; writing—original draft preparation, B.J.; writing—review and editing, B.J. and F.X.; visualization, B.J.; supervision, F.X.; funding acquisition, F.X. All authors have read and agreed to the published version of the manuscript.

**Funding:** This research was funded by NASA, grant numbers NNX14AK17G and NNX15AQ17G.

**Acknowledgments:** Special thanks go to the program managers Ramesh Kakar and Lucia Tsaoussi for support of the project. Additional thanks are necessary for Chuntao Liu, Thomas Winning, and Kevin Nelson for many helpful discussions about the research.

**Conflicts of Interest:** The authors declare no conflict of interest and the funders had no role in the design of the study; in the collection, analyses, or interpretation of data; in the writing of the manuscript, or in the decision to publish the results.

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
