# Peer review of "Characterizing Extratropical Tropopause Bimodality and its Relationship to the Occurrence of Double Tropopauses Using COSMIC GPS Radio Occultation Observations"

_remotesensing, doi:10.3390/rs12071109_

Round 1
Reviewer 1 Report
See attached file.

Reviewer 2 Report
The authors explore extratropical tropopause bimodality connections to the occurrence of double tropopauses using observations. The literature review is well documented, and the methodology is clear. The results are clear and discussed in detail. My comments (minor) are as follows:
- Lines 135: What is the reason for the decrease in the number of samples in 2017?
- Lines 153: The interpolation may introduce errors? Has this been quantified?
- The method section can be simplified using a flow chart. It will also be easier to illustrate the steps that may introduce errors to the final results.
- There are no statistical tests conducted to confirm bimodality. This is crucial to substantiate the main findings of the paper.
- Figure 6: Is there any physical reasoning for the occurrence of double tropopause observed near the ITCZ from 0-120E? Statistical significance of these results would have been useful too.
